# Establishment of a Temperature-Sensitive Model of Oncogene-Induced Senescence in Angiosarcoma Cells

**DOI:** 10.3390/cancers12020395

**Published:** 2020-02-08

**Authors:** Adilson da Costa, Michael Y. Bonner, Shikha Rao, Linda Gilbert, Maiko Sasaki, Justin Elsey, Jamie MacKelfresh, Jack L. Arbiser

**Affiliations:** 1Department of Dermatology, Emory University School of Medicine, Atlanta, GA 30322, USA; adilson_Costa@hotmail.com (A.d.C.); srao30@emory.edu (S.R.); lcgilbert@yahoo.com (L.G.); jpbower@emory.edu (J.M.); 2Department of Post-Graduate Studies, Instituto de Assistência Médica ao Servidor Público Estadual, Sao Paulo, SP 04029-000, Brazil; 3Department of Medical Biochemistry & Biophysics, Karolinska Institutet, 17177 Solna, Sweden; Michael.Bonner@ki.se; 4Atlanta Veterans Affairs Medical Center, Decatur, GA 30033, USA; mpapke@emory.edu; 5Department of Pathology, Emory University School of Medicine, Atlanta, GA 30322, USA; 6Winship Cancer Institute of Emory University, Atlanta, GA 30322, USA

**Keywords:** driver mutations, angiosarcoma, vascular proliferation, oncogenic Hras

## Abstract

Lesions with driver mutations, including atypical nevi and seborrheic keratoses, are very common in dermatology, and are prone to senescence. The molecular events that prevent senescent lesions from becoming malignant are not well understood. We have developed a model of vascular proliferation using a temperature-sensitive, large T antigen and oncogenic HRas. By elevating the temperature to 39 °C, we can turn off large T antigen and study the molecular events in cells with the Ras driver mutation. To assess the signaling events associated with the switch from a proliferative to a nonproliferative state in the constant presence of a driver oncogene, SVR cells were cultivated for 24 and 48 h and compared with SVR cells at 37 °C. Cells were evaluated by Western Blot (WB) gene chip microarray (GC) and quantitative reverse transcription polymerase chain reaction (RT-qPCR). Upon evaluation, a novel phenotype was observed in endothelial cells after switching off the large T antigen. This phenotype was characterized by Notch activation, downregulation of p38 phosphorylation, downregulation of the master immune switch IRF7, and downregulation of hnRNP A0. Switching off proliferative signaling may result in immune privilege and Notch activation, which may account, in part, for the survival of common skin lesions.

## 1. Introduction

Driver oncogenes are a common cause of both benign and malignant disorders of the skin. Examples of this include congenital nevi, which rarely undergo malignant transformation, and atypical nevi, which undergo transformation at a relatively slow rate. These lesions contain mutations in Ras and BRAF, respectively [1,2,3]. Seborrheic keratoses are among the most common lesions of the skin, and contain driver mutations in phosphoinositol-3 kinase and fibroblast growth factor receptor 3 [4,5,6]. Vascular malformations contain driver mutations in GNAQ, HRas and phosphoinositol-3 kinase subunits, underlying Sturge-Weber disease, and other vascular malformations [7]. In the presence of additional mutations, such as in tumor suppressors like p53 and p16ink4a, driver mutations lead to malignancy, but in lesions in which cell cycle controls are intact, driver mutations lead to indolent lesions in patients [8,9,10]. Introduction of driver oncogenes into normal cells leads to an eventual permanent cessation of cell growth, known as oncogene-induced senescence [8,9,10]. Greater knowledge of the events associated with the cessation of cell growth due to driver mutations would be of benefit in two ways. First, greater knowledge would make it possible to convert malignant tumors to benign tumors by interfering with signaling. Second, benign tumors cause significant morbidity to patients, and methods that can selectively destroy these lesions would be of benefit. Currently, the only treatment of these lesions are destructive, including surgery and other destructive modalities.

Oncogene-induced senescence (OIS) is difficult to study in vitro and in vivo. In vitro, the introduction of an oncogene into primary cells results in senescence within a short period of time, making it difficult to study. The grafting of benign lesions to immunocompromised mice was unsuccessful, as were attempts to adapt cells from benign lesions to tissue culture due to rapid senescence after plating. In order to overcome these difficulties, we have created a versatile model of oncogene-induced signaling in the SVR endothelial system, which contains a temperature-sensitive large T antigen and oncogenic HRas [10]. At 37 °C, the large T antigen is functional and overrides normal cell cycle restraints and these cells form tumors upon injection into animals. Switching the temperature to 39 °C turns off the large T antigen, but Ras is still active. We analyzed SVR cells at 37 °C vs. 39 °C and noted a novel phenotype upon switching the temperature. Among our findings are potential mitochondrial and immune evasive adaptations to OIS. Our findings may lead to insights into lesions which are benign from a cancer standpoint, but which may be highly morbid to the patient (driver mutation-induced vascular malformations, bathing trunk nevi, etc.).

## 2. Methods

### 2.1. Cells

Murine MS1 and SVR were cultured at 37 °C with 5% CO_2_, in Dulbecco’s modified Eagle’s medium (Sigma-Aldrich Inc., Saint Louis, MO, USA), enriched with 10% fetal bovine serum (Atlanta Biologicals, Lawrenceville, GA, USA), 1% complex of antibiotics-L-glutamine (10,000 IU/mL penicillin, 10,000 μg/mL streptomycin, and 29.2 mg/mL; Mediatech Inc., Manassas, VA, USA), and 1.5% L-glutamine 200 mM (Sigma-Aldrich Inc., Saint Louis, MO, USA), changed every 24 h [11]. MS1 and SVR were we incubated at either 37 °C or 39 °C for 24 h or 48 h. MS1 cells were murine pancreatic microvascular endothelial cells immortalized with a temperature-sensitive large T antigen, and SVR was derived from MS1 through the introduction of oncogenic H-ras.

### 2.2. Gene Microarray (GC)

In order to determine the genomic changes that occur when the large T proliferative signal is reversed, we performed a gene microarray on SVR cells with differing exposures, processed according to the Affymetrix^®^ WT Sense Plus (Thermo Fisher Scientific, Waltham, MA, USA) protocol [11]. The data analyses were done by using Partek Genome Studio version 6.6 (Partek, Inc., St. Louis, MO). The data were normalized by multi-array average (RMA) and the samples were grouped according to SVR, SVR24, and SVR48. An ANOVA test was performed to determine significant differential expression of genes among the groups. 

### 2.3. Quantitative Reverse Transcription Polymerase Chain Reaction (RT-qPCR)

Because the gene array may lead to false positives or negatives, we further confirmed our findings with qRT-PCR and Western blot. We focused on candidates involved in transcription, i.e., mitochondrial metabolism and MAP kinase signaling, as these have been implicated in OIS. Gene chip analysis results were confirmed via qRT-PCR using the Applied Biosystems 7500 FAST Real Time PCR System. Briefly, cDNA was generated from RNA extracts using SuperScript VILO cDNA Synthesis Kit (Invitrogen) and the Eppendorf Mastercycler gradient. Taqman primers for Notch1, IRF-7, p38 (MAPK11, MAPK12, MAPK13, and MAPK14), and S18 (endogenous control) were used with TaqMan Fast Universal PCR Mastermix (2X) (Applied Biosystems).

### 2.4. Western Blot (WB)

MS1 and SVR cells were lysed with 25% RIPA buffer containing halt protease and phosphatase single-use cocktail inhibitors (Thermo Fisher Scientific, Rockford, USA). Lysates were denatured with 5% β-mercaptoethanol in NuPage LDS buffer (Novex) and heated to 95 °C for five minutes. Proteins were run on 10% Bis-Tris gels (Novex) and transferred to polyvinylidene difluoride (PVDF) membranes, incubated at room temperature for one hour in 5% nonfat dry milk in tri-buffered saline and Tween 20 (Bio-Rad Laboratories, Inc., Hercules, CA, USA) (TBST) and probed in 5% nonfat dry milk in tri-buffered saline (TBS) at 4 °C overnight with antibodies: Akt (dilution 1:1000), cNotch1 (dilution 1:1000), FOXA3A (dilution 1:500), hnRNP A0 (dilution 1:250), Notch1 (dilution 1:1000), p38 (dilution 1:1000), p-p38 (dilution 1:1000), PS1 (dilution 1:250), PS2 (dilution 1:500), Sirt1 (dilution 1:500) and STAT-1 (dilution 1:250) from Cell Signaling Technologies, Inc. (Danvers, MA, USA); and FOXM1 (dilution 1:500), IFN-α (dilution 1:500), IRF-7 (dilution 1:2000), MEK3 (dilution 1:1000), MEK4 (dilution 1:250), CDK2a/p16INK4a (dilution 1:1000), Lamin B (dilution 1:1000), and MEK6 (dilution 1:250) from Abcam Inc. (Cambridge, MA, USA). For mitochondrial marker Western blotting, cells were lysed in denaturing lysis buffer (ThermoFisher NP0091) and homogenized through columns (OmegaBiotek, Norcross, GA, USA; Cat. #: HCR0001). PAGE and Western blotting were performed as described above. Mitochondrial marker antibodies (Cell Signaling Technology, 8674T), phosphor-p53 S15 (Abcam; cat. #1431) and phospho BAD S112 (Abcam; cat.# 129192) primary antibodies were diluted at 1:1000 and the blots were probed overnight at 4 °C. Membranes were then washed three times with TBST and incubated at room temperature for one hour with the corresponding antirabbit secondary antibody (Cell Signaling Technologies, Inc., Danvers, MA, USA). SuperSignal West Dura (Thermo Fisher Scientific, Rockford, IL, USA) was used to activate the horseradish peroxidase (HRP)-linked secondary antibodies. Image detection was achieved using the PXi 6 (Syngene USA, Frederick, MD, USA). 

## 3. Results 

We initially studied the expression of candidate signaling pathways regulated by the shutdown of proliferation by a shift from 37 °C to 39 °C. In addition, these pathways have been implicated in angiosarcoma, which has driver oncogene mutations. These include Notch1 and upstream MAP kinases, including MAPK 11, 12, 13, 14, MEK4, MEK6, and IRF7. Since we regard gene arrays as a method for generating hypotheses, signaling pathways were further confirmed by RT-qPCR (Figure 1). Since these proteins undergo posttranslational modifications, including proteolytic cleavage and phosphorylation, we confirmed expression by Western blot analysis as well. The top 100 up and downregulated genes are listed in Appendix A.

We then examined p38 MAP kinase signaling. p38 MAP kinase is widely implicated in mediation or prevention of apoptosis. Transcripts of p38 were modestly increased by temperature shift. However, levels of phosphorylated p38 (active p38) appeared to decrease over time, as did the upstream p38 activators, MEK4 and MEK6. Finally, levels of hnRNP A0, a p38 target, were decreased with increasing exposure to 39 °C. This indicates a functional decrease in p38 activation (Figure 2).

The master regulatory switch IRF7 also appeared to be regulated by temperature shift. While RT-qPCR appeared to show modest initial reduction of IRF7, western blot analysis gene chip analysis showed a coordinate downregulation of IRF7 and STAT1 (Figure 3).

A gene chip analysis revealed that Notch1 was modestly increased by temperature shift to 39 °C. Given the involvement of Notch upregulation and human cancer, we investigated the expression on the protein level of Notch1 and processing proteins. Western blot analysis revealed Notch1 cleavage. The induction of proteins involved in Notch1 cleavage, i.e., Presenilins 1 and 2, were noted at 24 h at 39 °C (Figure 4). Densitometry of Western blots is in Appendix A.

Finally, the CDK2a/p16INK4a senescence marker was shown to increase with time, with SVR cells at 39 °C showed even higher upregulation after 48 h of incubation compared to 24 h. The 39 °C induced senescence was noted after transferring cells to SVR at 37 °C after both the 24 and 48 h scenarios. Such senescence is confirmed when Lamin B is downregulated when SVR cells are placed at 39 °C, which irreversible.

The presence of oncogenic ras and temperature changes caused alterations in the expression of mitochondrial proteins. Notably, phosphorylation of BAD at Serine 112 was noted in SVR compared with MS1, and was decreased in SVR cells upon shifting to the nonpermissive temperature (Figure 5). This suggests that BAD phosphorylation (and inactivation) is required to relieve oncogene-induced stress, and decreased phosphorylation may mediate some cell death when cells are shifted to the nonpermissive temperature. p53 is dephosphorylated in SVR cells upon shifting to the nonpermissive temperature in ras transformed SVR cells, but not in MS1 cells, suggesting that this is not a nonspecific heat shock event, but that it is induced by oncogenic ras. The mitochondrial enzyme SDHA was decreased by ras transformation and further decreased by a shift to the nonpermissive temperature in ras-transformed SVR cells. This downregulation is more notable, given that there is more protein loaded in the temperature shifted 39 °C cells, but far less expression of SDHA, pBAD S112, and p153S15. The loss of SDHA has been shown to be an oncogenic event, manifested by both benign and malignant tumors [12,13,14].

## 4. Discussion and Conclusions

Benign tumors with driver mutations are common in the skin and, likely, also in other organs. These include atypical nevi with BRAF mutations, congenital nevi (HRas), seborrheic keratoses (phosphoinositol-3 kinase, FGFR3), blue nevi (GNAQ), and, potentially, other lesions like cherry angioma [1,2,3,4,5,6]. Malignant transformation occurs at a low rate in atypical nevi [15], and the transformation of the other lesions to malignancy is exceedingly rare. In addition, these lesions contain mutant proteins, but are rarely rejected by the immune system. To date, there is no medical therapy, including chemotherapy, that leads to the elimination of these lesions. Thus, the mechanisms of how these lesions remain quiescent or senescent is of great interest, as these oncogenes, which are the drivers of these common mutations, are also the drivers of aggressive malignancies, including melanoma [16], pancreatic carcinoma, multiple myeloma [17], and ovarian carcinoma [18], among others. Knowledge of the molecular events associated with oncogene-induced senescence may help convert malignant cells into quiescent or senescent cells.

We have previously developed a model of proliferative angiogenesis by sequentially introducing a temperature-sensitive SV40 large T antigen followed by oncogenic HRas into endothelial cells [10,11]. Cells containing both oncogenes develop into aggressive angiosarcoma [10] and have served as a useful and predictive preclinical model for chemotherapy and immunotherapy. The introduction of driver oncogenes into normal cells results in a brief burst of proliferation, followed by quiescence or senescence [19]. These cells exit the cell cycle, but can exist for long periods of time as living, but not replicative, cells [7,8,9]. The absence of proliferation makes these cells hard to study, because of the inability to expand them, and the rapid senescence that occurs upon introduction into culture. In order to better study this phenomenon, we utilized the temperature-sensitive nature of the cells and turned off the large T antigen by switching the cells to 39 °C. The major advantage of our system is that a large number of cells can be grown and then shifted to nonpermissive temperatures. Other models of OIS are limited by poor growth after a few days and, thus it is difficult to obtain the quantities of cells required to do the analyses that we have done. Also, given that our system uses a driver oncogene in endothelial cells, it may provide valuable data for the study of human vascular malformations, the majority of which also contain driver mutations in the endothelial cells.

Using gene array analysis, we studied the signaling changes that occur at 24 and 48 h after the temperature switch to 39 °C, with confirmation of candidates by RT-PCR and Western blot analysis. It appears that irreversible change occurs at 24 h. Our initial findings suggest that there is a generalized downregulation of upstream activators of p38 upon shifting to the nonpermissive temperature. This suggests that the maintenance of the senescent phenotype results in a low level of p38 signaling and that p38 activators might be effective in mediating senolytic therapies. Consistent with the downregulation of p38 activators, we found downregulation of hnRNP A0 at 48 h after temperature shift. These RNA changes were confirmed at the protein level as well (Figure 2). Of interest, the p38 MAP kinase interacts with the hnRNP A0 protein in order to stabilize the GADD45a RNA from rapid degradation. GADD45a is required for G2 arrest after tumor cells are exposed to DNA damaging agents such as doxorubicin [20], and interference with this pathway leads to increased sensitivity among these agents, in part by an inability to undergo cell cycle arrest after a genotoxic exposure. The downregulation of MEKK4 and MEKK6 observed upon temperature shift is consistent with p38 downregulation, as these are upstream kinases of p38. Of further interest, p38 MAPK is not expressed in Sturge-Weber lesions, while ERK and JNK are [21]. Therefore, both our system and Sturge-Weber syndrome consist of endothelial cells containing a driver oncogene, so the induction of p38 might be required to cause the regression of vascular malformations.

In addition to the effects on p38 signaling, we saw a downregulation of the master immune switch IRF-7 in temperature-shifted cells. IRF-7 is present in most cells [22,23,24], but inhibited in Kaposi’s sarcoma [24,25], down-regulated in bone-metastatic [26] breast cancer cells [27], and epigenetically silenced in lung cancer [28]. In most cells, IRF-7 is regulated by IFN/STAT-1 in a positive feedback loop [22,23], involving p38 MAPK [29]. Stat-1 and IFNα are also downregulated by the temperature shift (Figure 3), potentially conferring immune privilege to these senescent cells, and may explain why these lesions can exist without an immune response to the mutant driver oncogene. 

We noted initial downregulation of Notch1 in the gene array but, surprisingly, Notch activation was observed at the protein level by the temperature shift. The Notch C-terminal domain is important for Presenilin1 activity [30,31], since it regulates Notch1 signaling action by liberating Notch Intracellular Cytoplasmic domain (NICD) [31,32,33] in a negative feedback manner [33,34]. This negative feedback might explain the downregulation of Notch1 RNA along with the increase in Notch activation at the protein level (Figure 4). Yoshida et al. demonstrated the role of Notch1 signaling in the regulation of endothelial cell senescence. Notch1 overexpression was shown to prolong the lifespan of endothelial cells via suppression of p38 MAPK activity [35]. Of interest, Notch1 is expressed in atypical nevi compared to normal cells, while melanoma demonstrates expression of additional Notch genes [36].

We demonstrated that SVR cells undergo irreversible senescence upon temperature-shift (Figure 6). Induction of CDK2a/p16INK4a was evaluated as it is a marker of OIS rather than heat shock-induced stress [37,38,39,40,41,42]. Conversely, Lamin B loss is a well-established biomarker of cellular senescence [43]. In this study, we observe that CDK2a/p16INK4a was gradually time- and temperature-upregulated in SVR cells, with induction after 24 h at 39 °C, which was nonreversible upon shifting back to 37 °C. Further confirmation of this phenotype is demonstrated by Lamin B downregulation when SVR cells are placed at 39 °C without reversal after shifting back to 37 °C. These findings are consistent with OIS rather than heat shock-induced stress.

While both heat shock and OIS are causes of senescence, they are distinguished by important features. The induction of beta galactosidase is common to both, so this cannot distinguish the two inducers of senescence. However, heat shock does not induce the expression of p16ink4a, but OIS does [8,42]. We also observe the induction of p16ink4a in an irreversible manner and this is maintained after shifting back to 37 °C and thus is consistent with OIS rather than heat shock. Finally, the induction of senescence-associated beta galactosidase is not regarded as a reliable measure of oncogene-induced senescence as there is little expression in senescent nevi caused by Braf in humans [44].

Mitochondrial adaptations are required to allow cells to survive oncogene-induced stress. We examined the expression of a panel of mitochondrial-associated proteins in terms of response to oncogene-induced stress due to the removal of the large T antigen. Of note, the mitochondrial protein BAD is phosphorylated at Ser 112 only in the ras transformed SVR cells, compared with parental MS1 cells, and is decreased upon temperature shift to 39 °C (Figure 5). BAD is phosphorylated at Ser 112 which inactivates the mitochondrial apoptosis associated with BAD heterodimerization [45]. The increase in BAD phosphorylation suggests that it is required for alleviating the oncogene-induced stress. VDAC1 appears to be upregulated by temperature shift in both MS1 and SVR, with a more pronounced effect in the MS1 (Figure 5), implying that ras might downregulate VDAC1 under conditions of stress [44,45]. p53 is rapidly phosphorylated in SVR cells upon shift to 39 °C (Figure 5) and it is well known that phosphorylated p53 translocates to the mitochondria and may mediate apoptosis [46]. The loss of BAD phosphorylation upon temperature shift suggests that the coordinated action of p53 phosphorylation and BAD dephosphorylation may mediate mitochondrial-mediated, oncogene-induced cell death. Stimuli that maintain BAD phosphorylation or decrease p53 phosphorylation may allow for long term survival of lesions with oncogene-induced stress. The loss of succinate dehydrogenase A (SDHA) protein is striking, because the loss of SDHA has been shown to be oncogenic, and metabolic defects caused by the accumulation of metabolites might be required to maintain benign lesions with driver mutations as well [14].

Our studies identified novel therapeutic targets for the treatment of benign lesions with driver mutations, such as vascular malformations. While histologically benign, these lesions can result in significant decrease in the quality of life of patients. Among the novel targets elucidated by our study, we demonstrate the activation of p38 MAP kinase, the reversal of immune privilege (upregulation of IRF7 and stat-1), and the reversal of metabolic abnormalities. Potential novel therapeutics include Sirt3 activators (i.e., honokiol) [47,48], which may be effective in eliminating cells with driver mutations by promoting SDH and p38 activation and thus converting OIS into cell death. 

## Figures and Tables

**Figure 1 cancers-12-00395-f001:**
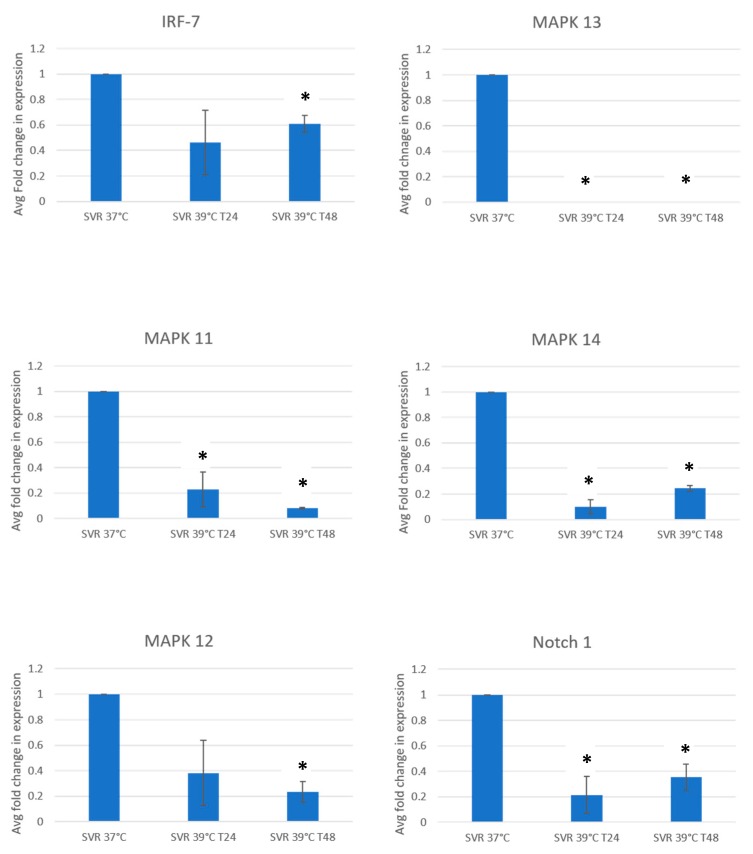
qPCR SVR cells under 37 °C and 39 °C (in 24 and 48 h) for evaluating Notch1, IRF-7, and p38 MAPK kinases. Experiments were repeated in threefold, and decreases were significant at *p* = 0.05 at 48 h marked by an asterisk (*).

**Figure 2 cancers-12-00395-f002:**
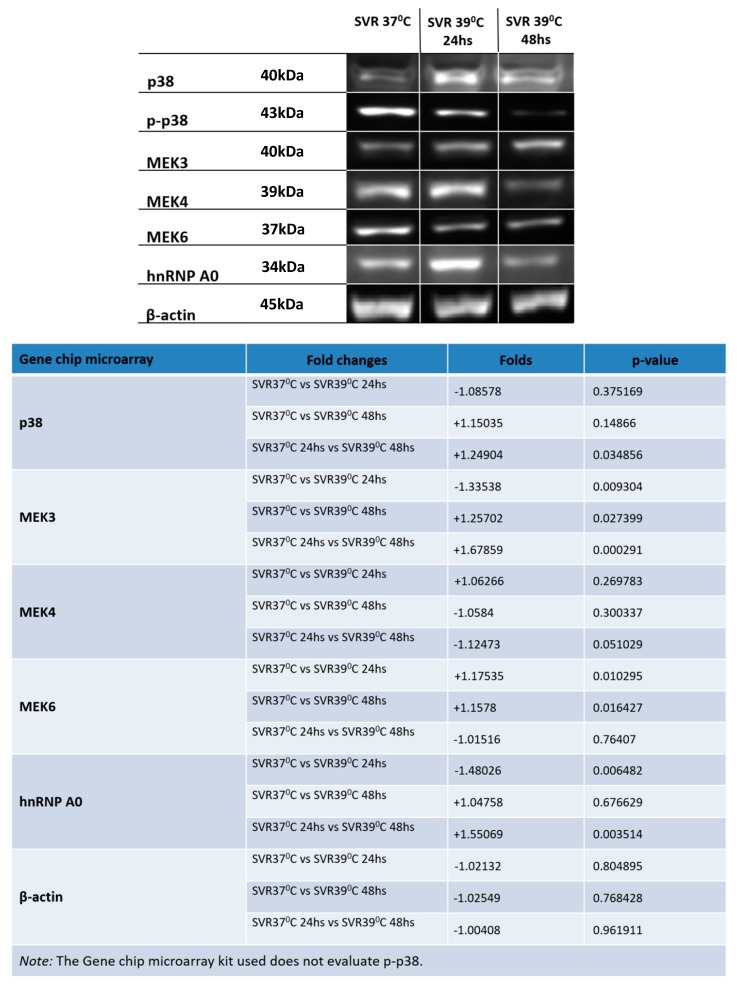
Western blot analysis conducted on SVR cells under 37 °C and 39 °C (in 24 and 48 h) for evaluating p38 and MAPK kinase kinase. Experiments were repeated in triplicate and *p*-values are displayed at the lower right panel.

**Figure 3 cancers-12-00395-f003:**
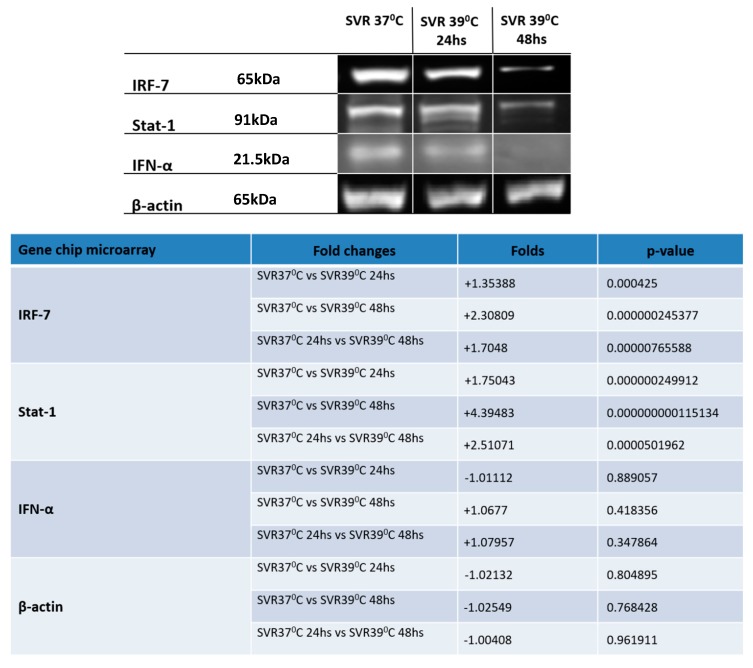
Western blot analysis conducted on SVR cells under 37 °C and 39 °C (in 24 and 48 h) for evaluating IRF-7 and molecules related to its pathway. Western blot analysis conducted on SVR cells under 37 °C and 39 °C (at 24 and 48 h) for evaluating Notch1 and Presenilin 1 and 2. Experiments were repeated in triplicate and *p*-values are displayed in the lower right panel.

**Figure 4 cancers-12-00395-f004:**
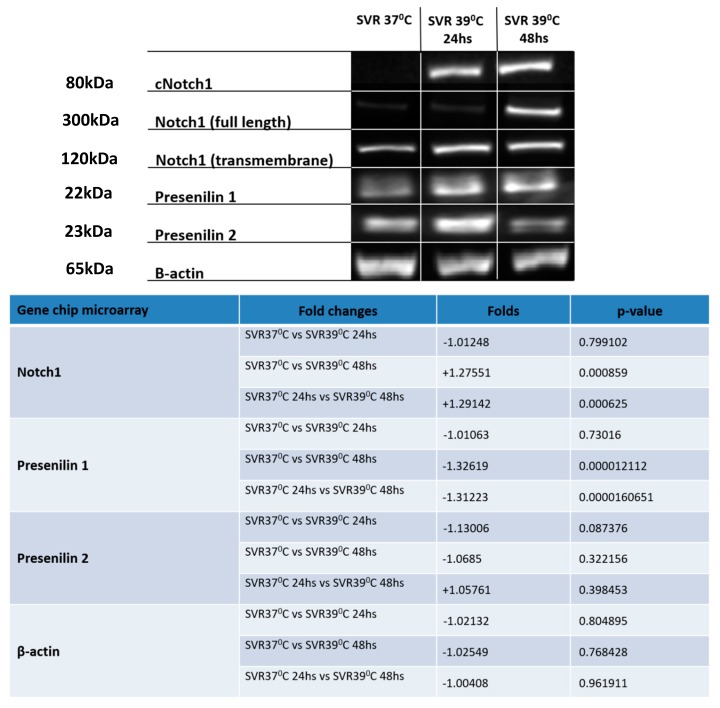
Western blot analysis conducted on SVR cells under 37 °C and 39 °C (at 24 and 48 h) for the evaluation of Notch1 and Presenilin 1 and 2. Experiments were repeated in triplicate and *p*-values are displayed in the lower right panel.

**Figure 5 cancers-12-00395-f005:**
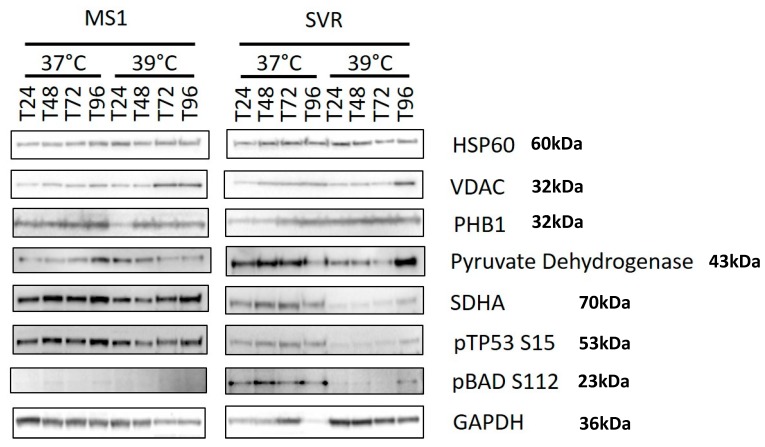
Western blot analysis conducted on MS1 and SVR cells under 37 °C and 39 °C for evaluating mitochondrial proteins. Temperature and hours of exposure are listed at the top of the representative Western blots. The identities of the proteins are given on the right.

**Figure 6 cancers-12-00395-f006:**
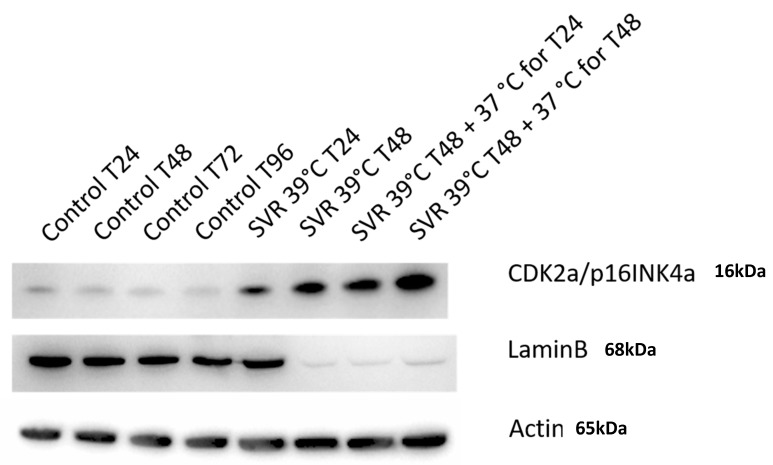
Western blot analysis conducted on SVR cells cultured at 37 °C and 39 °C (in 24 and 48 h), as in Figure 2, for evaluating a/p16INK4a and Lamin B. The continued elevation of p16ink4a after reversion to 37 °C suggests that the induction of senescence is irreversible.

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
