# Peer review of "Establishment of a Temperature-Sensitive Model of Oncogene-Induced Senescence in Angiosarcoma Cells"

_cancers, 2020, doi:10.3390/cancers12020395_

Round 1
Reviewer 1 Report
The authors attempt to open a new and novel window on understanding the activity of oncogenes, and they respond to robust changes, but they lack the comprehensible way to convey this message to readers as it will be written in the upcoming parts:
Major points:
The paper is written and structured very confusingly. Readers will not be able to understand the logic of the experiment and find the coherence of the sections.
-) Introduction was explained, very obvious observations until the reader reaches this part: "We have created a versatile model of oncogene-induced signaling ...". From here on, the authors begin to explain what they have done, without having conveyed knowledge or raised the question/problem in the earlier sections of the introduction. In the introduction the reader expects to read a short history of the field, then the question should be raised how others have tried to solve it and what is the proposed solution of the current paper? This should be seen as a step-by-step guide.
-) The same applies to the results section. The reader needs to extract the understanding of each result section before moving on to the next, and the coherence between the sections should be presented.
For example, the authors reported on the changes or non-difference status of some proteins such as stat1, vdac1, sdha etc...without explaining their relation to the whole picture. If showing these proteins is intended to convey any messages, then it should be written in the text, if not, what is the purpose of showing these proteins?
It is understandable that authors have interesting findings and want to share them with others, but it should be so that the consistency of the paper is not affected.
-) The section on results must be more informative and includes the reason for conducting each experiment, the result obtained and how it fits into the big picture.
-) To confirm the data obtained from the in vitro part, it is strongly recommended to perform an in vivo experiment. Many factors could alter the in vitro results while these could be eliminated in vivo experiments.
Minor points:
Abbreviated words must be explained at least once in the text (in the first place where they were used)
Cell lines must be explained within 1-2 sentences.
Author Response
The paper is written and structured very confusingly. Readers will not be able to understand the logic of the experiment and find the coherence of the sections.
-) Introduction was explained, very obvious observations until the reader reaches this part: "We have created a versatile model of oncogene-induced signaling ...". From here on, the authors begin to explain what they have done, without having conveyed knowledge or raised the question/problem in the earlier sections of the introduction. In the introduction the reader expects to read a short history of the field, then the question should be raised how others have tried to solve it and what is the proposed solution of the current paper? This should be seen as a step-by-step guide.... This has been done in the revised manuscript
For example, the authors reported on the changes or non-difference status of some proteins such as stat1, vdac1, sdha etc...without explaining their relation to the whole picture. If showing these proteins is intended to convey any messages, then it should be written in the text, if not, what is the purpose of showing these proteins?... this has been done in the revised manuscript. We describe pertinent negatives so as to better describe the mitochondrial status of the cells.
To confirm the data obtained from the in vitro part, it is strongly recommended to perform an in vivo experiment. Many factors could alter the in vitro results while these could be eliminated in vivo experiments... We are writing a grant proposal to do this. One of the technical difficulties we have observed is the senescent cells are hard to remove from the plates by enzymatic digestion, which is commonly observed in other senescent cells.
Reviewer 2 Report
In this manuscript, authors have conducted experiments using SVR cell line containing oncogenic H-ras which are derived from MS1 murine pancreatic microvascular endothelial cells immortalized with a temperature sensitive large T antigen. Authors have analyzed SVR cells at 37°C vs. 39°C and observed a novel phenotype upon temperature switch with modulation in various signaling pathways. Authors conclude that the resulting phenotype due to temperature switch may be relevant to human benign lesions that contain mutations in oncogenes, yet evade immune detection and interventions that induce p38 MAP kinase signaling or inhibit Notch signaling may be effective in eliminating these lesions.
Overall, the manuscript presents some in vitro mechanistic data using a cell line derived from a parental MS1 murine pancreatic microvascular endothelial cells. There are various major concerns in the manuscript which are provided below.
Authors have written that “Gene chip analysis revealed Notch 1 was modestly increased by temperature shift to 39°C”. However, Fig 1 RT-qPCR data suggest a significant decrease in Notch 1. Authors should explain this. It appears that b-actin blots shown in figure 2, 3 and 4 are same. Authors have shown only one internal control blot for 14 other proteins of interest shown in these figures. Authors are suggested to provide separate internal control blots for each of the protein or at least in each of these figures. Authors have not shown western blot data for MAPK 11-14 which were drastically downregulated at gene level. Fig 6, Western blot data, was the GAPDH used as a loading control. If yes, then it seems loading was not equal as the GAPDH expression is highly variable in samples. If not, authors need to provide blot for loading control, possibly for each of the protein shown in the fig 6. Only one of the figures (Fig 6) contain data from MS1 cells, rest all the data is shown for single cell line SVR. Authors need to replicate these results for at least one more cell line to conclude these data. Authors have not provided Gene array data in the manuscript. They may provide this data as supplementary figure. Authors should carefully proof read the manuscript for typing and grammatical errors. Ex. Legend to Fig 2, Western blot analysis conducted on SVR cells under 37°C and 337°C and 39°C 9°C. Please correct the typing errors in this sentence.Author Response
Authors have written that “Gene chip analysis revealed Notch 1 was modestly increased by temperature shift to 39°C”. However, Fig 1 RT-qPCR data suggest a significant decrease in Notch 1. Authors should explain this... this has been done in the revised paper. In general, we find that RT-qPCR is more accurate than gene chip and we confirm our RT-qPCR findings with additional studies
Authors have not shown western blot data for MAPK 11-14 which were drastically downregulated at gene level.... We had difficulty with the commercially available antibodies on Western blot. We would like to address the role MAPK 11-14 in overexpression and siRNA studies in a future manuscript. Western blot of other upstream kinases is demonstrated in the revised figure 2
Only one of the figures (Fig 6) contain data from MS1 cells, rest all the data is shown for single cell line SVR. Authors need to replicate these results for at least one more cell line to conclude these data. ... other cell lines do not contain the temperature sensitive large T and ras together.
Authors have not provided Gene array data in the manuscript. They may provide this data as supplementary figure.... This has been done in the revised manuscript
Authors should carefully proof read the manuscript for typing and grammatical errors. Ex. Legend to Fig 2, Western blot analysis conducted on SVR cells under 37°C and 337°C and 39°C 9°C. Please correct the typing errors in this sentence. ... This has been done in the revised manuscript
If yes, then it seems loading was not equal as the GAPDH expression is highly variable in samples.... The loading in fig 6 of GADPH is not fully even. However, it demonstrates some large differrences, as loss of pBADS112, p53S15, and SDHA at the 39C temp shift in SVR, when GAPDH is more heavily loaded.
Reviewer 3 Report
The authors used the MS1/SVR cellular system as a model of oncogene induced senescence (OIS). MS1 is an immortalized endothelial cell line whereas SVR cells are MS1 cells transformed with oncogenic HRAS. By switching temperature to 39°C, the authors claim that SVR cells go into OIS. Using Western Blot (WB) gene chip microarray (GC) and quantitative reverse transcription polymerase, they observed Notch activation, downregulation of p38 phosphorylation, downregulation of the master immune switch IRF7, and downregulation of hnRNA0 at 39°C.
The MS1 and SVR cellular system is a valuable tool for experiments on human angiosarcoma but the paper does not demonstrate its interest for studying OIS. Although the SVR cells express p16INK4A at 39°C, the authors do not show evidence of OIS such as senescence-associated beta-galactosidase activity or senescence-associated heterochromatin foci. We can therefore assume that the effects observed are just linked to the stress induced by the heat shock. To further confirm that the effects are linked to oncogenic HRAS, they also need to show that when MS1 (lacking HRAS) are shifted to 39°C they do not go into OIS and do not activate/inhibit the same pathways than SVR.
Signal transduction pathways analysed in this paper (p38, MEK, STAT) are regulated at the protein level by phosphorylation/dephosphorylation. The study of mRNA levels for these proteins is therefore not relevant. Indeed, opposite results are sometimes found between GM and WB. For example, STAT1 displays a strong increase (+4,3) in GM but clear decrease in WB (Figure 4). The authors need to analyse protein phosphorylation for p38, MEK3, MEK4, MEK6 and STAT1 and quantify for each protein the ratio phosphorylated protein/total protein.
Figure 1: indicate above each graph whether it is significant.
Figure 2: There is a mistake in the legend: "under 37°C and 337°C and 39°C 9°C"!
Figure 2, 3 and 4: western blots need to be quantified.
Figure 6: The levels of total BAD and total p53 need to be shown. The GAPGH blots shows strong differences between the lanes suggesting different quantities of proteins.
Reference 11 is missing
Author Response
OIS such as senescence-associated beta-galactosidase activity or senescence-associated heterochromatin foci. ...these are not specific for oncogene induced senescence and can be activated by heat shock. On the other hand, p16 is not induced by heat shock induced senescence and therefore the induction of p16 by temperature shift is highly suggestive of oncogene induced senescence
Nucleic Acids Res. 2015 Jul 27; 43(13): 6309–6320. Published online 2015 Jun 1. doi: 10.1093/nar/gkv573 PMCID: PMC4513856 PMID: 26032771
Mechanism of heat stress-induced cellular senescence elucidates the exclusive vulnerability of early S-phase cells to mild genotoxic stress
Artem K. Velichko,1,† Nadezhda V. Petrova,1,2,† Sergey V. Razin,1,2,3,* and Omar L. Kantidze1,3,* r
In addition, lamin B is a heat shock protein (see ref below). If senescence would be due to heat shock, then lamin B should be induced by the temperature shift instead of reduced as we observe in Fig 5
J Cell Physiol. 1999 Jan;178(1):28-34.
Lamin B is a prompt heat shock protein.
Dynlacht JR1, Story MD, Zhu WG, Danner J.
Author information
Figure 2: There is a mistake in the legend: "under 37°C and 337°C and 39°C 9°C"!... this has been corrected in the revised paper
Round 2
Reviewer 1 Report
authors need to show the original blots of WBs, for example in figures 3 and 4 it seems those blots are manipulated, there is no need to draw a box around each lane. in all WBs the protein size should be written. in fig 6 there is no need to write a, b,c d, beside the blots, it makes the figure unnecessary crowded. Instead of those letters, the author can write the size of proteins. As it is impossible to run all WBs on the same membrane, authors need to show the individual gapdh control for each blot that has been separately performed.
Author Response
for example in figures 3 and 4 it seems those blots are manipulated, there is no need to draw a box around each lane. in all WBs the protein size should be written. in fig 6 there is no need to write a, b,c d, beside the blots, it makes the figure unnecessary crowded. Instead of those letters, the author can write the size of proteins... we have done this in the revised manuscript.
As it is impossible to run all WBs on the same membrane, authors need to show the individual gapdh control for each blot that has been separately performed.... our use of a common loading standard on protein is generally accepted in the literature
Reviewer 2 Report
Authors are suggested to quantify western blot data and present as bar diagram with error bars and significance level If the western blots are performed in triplicates.
Author Response
Authors are suggested to quantify western blot data and present as bar diagram with error bars and significance level If the western blots are performed in triplicates. ... Western blots were performed in duplicates so we did not use error bars. RT-PCR data in in triplicates and the significance is indicated as part of the figures.
Reviewer 3 Report
The authors have not addressed my main concern about lack of proof of senescence in their cells. Moreover, they have only answered half of my questions/comments. They have not answered the following comments:
Figure 2, 3 and 4: western blots have not been quantified.
Figure 6: The levels of total BAD and total p53 are still not shown, this is particularly important as the GAPGH blots shows strong differences between the lanes suggesting different quantities of proteins.
Reference 11 on MS1 and SVR cells is still missing
Author Response
The authors have not addressed my main concern about lack of proof of senescence in their cells... the use of senescence associated b-gal as a marker is being decreased because of lack of specificity for senescence. Even the Campisi lab, which originated the senescence associated b-gal assay, is no longer universally using it -see paper below. At this point, induction of p16ink4a and loss of lamin B are well accepted markers of senescence. In addition, lack of expression has been observed in nevi with the oncogene Braf (see second paper below)
JCI Insight. 2019 Dec 19;4(24). pii: 130056. doi: 10.1172/jci.insight.130056.
Secretion of leukotrienes by senescent lung fibroblasts promotes pulmonary fibrosis.
Wiley CD1, Brumwell AN2, Davis SS1, Jackson JR2, Valdovinos A1, Calhoun C3, Alimirah F1, Castellanos CA2, Ruan R2, Wei Y2, Chapman HA2, Ramanathan A1,4, Campisi J1,5, Jourdan Le Saux C2,3.
Author information
Abstract
Accumulation of senescent cells is associated with the progression of pulmonary fibrosis, but mechanisms accounting for this linkage are not well understood. To explore this issue, we investigated whether a class of biologically active profibrotic lipids, the leukotrienes (LT), is part of the senescence-associated secretory phenotype. The analysis of conditioned medium (CM), lipid extracts, and gene expression of LT biosynthesis enzymes revealed that senescent cells secreted LT, regardless of the origin of the cells or the modality of senescence induction. The synthesis of LT was biphasic and followed by antifibrotic prostaglandin (PG) secretion. The LT-rich CM of senescent lung fibroblasts (IMR-90) induced profibrotic signaling in naive fibroblasts, which were abrogated by inhibitors of ALOX5, the principal enzyme in LT biosynthesis. The bleomycin-induced expression of genes encoding LT and PG synthases, level of cysteinyl LT in the bronchoalveolar lavage, and overall fibrosis were reduced upon senescent cell removal either in a genetic mouse model or after senolytic treatment. Quantification of ALOX5+ cells in lung explants obtained from idiopathic pulmonary fibrosis (IPF) patients indicated that half of these cells were also senescent (p16Ink4a+). Unlike human fibroblasts from unused donor lungs made senescent by irradiation, senescent IPF fibroblasts secreted LTs but failed to synthesize PGs. This study demonstrates for the first time to our knowledge that senescent cells secrete functional LTs, significantly contributing to the LT pool known to cause or exacerbate IPF
J Invest Dermatol. 2007 Oct; 127(10): 2469–2471. Published online 2007 May 24. doi: 10.1038/sj.jid.5700903 PMCID: PMC2292406 NIHMSID: NIHMS44273 PMID: 17522702
Absence of senescence-associated β-galactosidase activity in human melanocytic nevi in vivo
Figure 2, 3 and 4: western blots have not been quantified... since these were done in duplicate, we did not feel that it was appropriate to use error bars
Figure 6: The levels of total BAD and total p53 are still not shown, this is particularly important as the GAPGH blots shows strong differences between the lanes suggesting different quantities of proteins....total p53 was very faint. The relative overloading of GAPDH in the SVR 39oC with greatly reduced levels of p53, SDH graphically emphasizes the changes that we have observed.
Reference 11 on MS1 and SVR cells is still missing... this has been added in the revised manuscript